# Analysis of Let-7 Family miRNA in Plasma as Potential Predictive Biomarkers of Diagnosis for Papillary Thyroid Cancer

**DOI:** 10.3390/diagnostics10030130

**Published:** 2020-02-28

**Authors:** Ewelina Perdas, Robert Stawski, Krzysztof Kaczka, Maria Zubrzycka

**Affiliations:** 1Department of Cardiovascular Physiology, Faculty of Medicine, Medical University of Lodz, 92-215 Lodz, Poland; ewelina.perdas@umed.lodz.pl (E.P.);; 2Department of Clinical Physiology, Faculty of Medicine, Medical University of Lodz, 92-215 Lodz, Poland; 3Department of General and Oncological Surgery, Medical University of Lodz, 92-213 Lodz, Poland; krzysztofkaczka@poczta.fm

**Keywords:** PTC, miRNA, let-7, ddPCR, diagnostics

## Abstract

The most common histological type of thyroid cancer is papillary thyroid carcinoma (PTC). Radical resection of the thyroid gland is currently the recommended method of treatment. Almost 75% of thyroidectomies performed just for diagnostic purposes are benign. Thus, the confirmation of innovative and more precise noninvasive biomarkers holds promise for the detection of PTC, which may decrease the number of unnecessary thyroid lobectomies. In this work, using the droplet digital PCR (ddPCR) method, we have analyzed the level of five miRNAs (let-7a, let-7c, let-7d, let-7f, and let-7i) in the plasma of patients with PTC and compared them with those of a healthy control group to investigate whether miRNAs also have value in the management of PTC. Levels of four miRNAs, namely let-7a, let-7c, let-7d, and let-7f, were significantly higher in PTC patients than healthy controls. Thus, the analysis of circulating let-7 can be a useful tool and support the currently used methods for PTC diagnosis. However, our observation requires further research on a larger patient group.

## 1. Introduction

Over the recent decades, there has been an increase in the incidence of thyroid cancer worldwide [1]. Papillary thyroid carcinoma (PTC) is the most common histological type of thyroid cancer. Between 1970 and 2010, the proportion of papillary thyroid cancers increased from 58.0% to 85.9%. At the same time, the number of total thyroidectomies, which are currently the recommended method of treatment, increased 2.5 times [1,2]. Almost 75% of thyroidectomies performed just for diagnostic purposes are benign [3,4]. Thus, the confirmation of innovative and more precise noninvasive biomarkers holds promise for the detection of PTC, which may decrease the number of unnecessary thyroid lobectomies and prevent the use of thyroid replacement therapy for the rest of a patient’s life.

Expression of circulating microRNAs (miRNAs) has been found as potential blood-based biomarker in the diagnosis of cancer [5]. The miRNAs are short noncoding single-stranded post-transcriptional regulators of gene expression. Normal tissue and cancers are different in the expression profiles of circulating miRNAs. The exact mechanism by which miRNAs are released into the body fluids still remains unclear, but various mechanisms seem to be involved. The miRNAs could be passively leaked or actively secreted into the circulation. Passive release into body fluids as a result of cell death by tissue degradation is associated with malignancy. The miRNAs can also be merged in exosomes or loaded into microvesicles through membrane budding and then released in the extracellular space. The miRNAs of exosome origin have been described to represent a subdivision of about 3% of the total amount of cell-free miRNAs [6]. A large number of miRNAs is released through an RNA-binding protein (e.g., high density lipoprotein (HDL) and nucleophosmin 1 (Npm1)) resulting in the stabilization of miRNA [7,8].

The differential expression of miRNAs in the circulation of patients with cancer when compared with healthy controls has been documented by numerous studies, making miRNAs a promising noninvasive biomarker. This makes them hopeful diagnostic and therapeutic targets in deregulated pathways in cancer development. Deregulation of the let-7 family is widely explored in many cancer types and is suggested to be a valuable diagnostic tool in liquid biopsy including lung cancer (let-7a, let-7c, let-7f), prostate cancer (let-7a), gastric cancer (let-7a, let-7c, let-7i, let-7f), breast cancer (let-7c, let-7b, let-7g), ovarian cancer (let-7b, let-7f, let-7i), hepatocellular carcinoma (let-7b, let-7f), acute myeloid leukemia (let-7b, let-7d), and colorectal cancer (let-7a). However, their potential as prognostic biomarkers has also been emphasized in several cancer types such as myelodysplasia (let-7a), lung cancer (let-7b, let-7f, let-7i), ovarian cancer (let-7f), hepatocellular carcinoma (let-7b, let-7f), multiple myeloma (let-7e), and prostate cancer (let-7d) [9]. Let-7 miRNA family is a promising research object in PTC. Nevertheless, there is limited literature on circulating let-7 in PTC. Expression of let-7a, let-7,c, let-7d, let-7f, and let-7g has not been described in plasma. For the examined miRNAs (let-7b, let-7e, and let-7i), overexpression in plasma was found [10].

Before the practical application, several challenges need to be overcome. Commonly applied relative quantification of circulating miRNAs by quantitative PCR (qPCR) is limited by the difficulty of finding reliable endogenous reference genes in the plasma or serum. Recently, droplet digital PCR (ddPCR) has been shown to overcome these methodical difficulties, allowing absolute quantification without the need for internal/external normalization. Moreover, the ddPCR technique has shown increased precision and sensitivity in detecting low target copies, and it is also relatively insensitive to potential PCR inhibitors, thus producing more reliable results than conventional qPCR [11].

In this study, using ddPCR, we have examined the level of five miRNAs (let-7a, let-7c, let-7d, let-7f, and let-7i) in the plasma of patients with PTC and compared them with those of a healthy control group to investigate whether miRNAs also have a value in the management of PTC. According to our knowledge, the expression of let-7a, let-7c, let-7d, and let-7f has never been tested in plasma of PTC patients.

## 2. Results

### 2.1. Demographics of the Study Groups

We compared the levels of five miRNAs (let-7a, let-7c, let-7d, let-7f, and let-7i) in plasma, by ddPCR, between the PTC cases and healthy subjects. The median age for cancer patients (*n* = 49) was 48 years, range 20–76. The median age of healthy control individuals (*n* = 21) was 47 years, range 29–67. The detailed data are presented in Table 1.

### 2.2. Plasma miRNA in PTC and Healthy Control Groups

The levels of four miRNAs (let-7a, let-7c, let-7d, let-7f) were significantly higher in PTC patients than healthy controls (Mann–Whitney: *p* = 0.036; *p* = 0.025; *p* = 0.01; *p* = 0.013, respectively). The level of let-7i was higher in PTC group compared to control, but the difference is statistically insignificant (Mann–Whitney: *p* = 0.059) (Table 2 and Figure 1).

No statistical differences in miRNA levels were found among men in the PTC group compared to controls (Mann–Whitney: *p* = 0.424; *p* = 0.505; *p* = 0.142; *p* = 0.689; *p* = 0.182). Similarly, no statistical differences in miRNA levels were found among women of both groups (Mann–Whitney: *p* = 0.247; *p* = 0.171; *p* = 0.231; *p* = 0.076; *p* = 0.318). Additionally, no statistical change was found in the PTC group itself (Mann–Whitney: *p* = 0.228; *p* = 0.181; *p* = 0.192; *p* = 0.17; *p* = 0.39) as well as in the control group (Mann–Whitney: *p* = 0.5; *p* = 0.2; *p* = 0.118; *p* = 0.594; *p* = 0.145) for all miRNAs (Table 3).

### 2.3. Tumor and Lymph Nodes Characteristics in PTC Group

The primary tumor (T) and regional lymph nodes (N) classification was performed according to the system established by the International Union Against Cancer (UICC, 2010).

There is no statistical difference in miRNA levels compared to T_1_ + T_2_ and T_3_ + T_4_ tumor stages (Mann–Whitney: *p* = 0.926; *p* = 0.926; *p* = 0.676; *p* = 0.553; *p* = 0.852). Moreover, no statistical difference was found between the group of patients with regional lymph node metastases (N_1_) compared to no regional lymph nodes metastases (N_0_) and the group where lymph nodes could not be assessed (N_x_) for all miRNAs (Kruskal–Wallis: *p* = 0.372; *p* = 0.451; *p* = 0.685; *p* = 0.685; *p* = 0.276) (Table 3, Figure 2 and Figure 3).

### 2.4. Receiver Operating Characteristic (ROC) Curve

Receiver operating characteristic (ROC) curve analyses were conducted to calculate the diagnostic value of the selected miRNAs (let-7a, let-7c, let-7d, let-7f) in the PTC compared to healthy control groups. The areas under the ROC curves (AUCs) were measured to estimate the specificity and sensitivity of miRNA to diagnose patients with PTC. The selection of an optimal cutoff point for discriminating between the PTC group and the healthy control group has been allowed (set) by the Youden index (J). J is the maximum vertical distance between the ROC curve and the diagonal reference line. Youden index is defined as J = maximum (sensitivity) + (specificity) − 1. The data are illustrated in Figure 4.

The levels of let-7a, let-7c, let-7d, and let-7f miRNAs in the PTC cases were significantly higher in comparison with the healthy controls. An optimal cutoff point for let-7a was indicated at 28.5 copies/µL with a sensitivity of 74% and a specificity of 38% (AUC = 0.66, *p* < 0.05, 95% confidence interval = 0.522–0.796); for let-7c, 50.5 copies/µL with a sensitivity 50.5% and a specificity of 14% (AUC = 0.67, *p* < 0.05, 95% confidence interval = 0.542–0.798); for let-7d, 6.8 copies/µL with a sensitivity 71% and a specificity of 33% (AUC = 0.70, *p* < 0.05, 95% confidence interval = 0.562–0.832); for let-7f, 16.2 copies/µL with a sensitivity of 59% and a specificity of 19% (AUC = 0.69, *p* < 0.05, 95% confidence interval = 0.556–0.823); and for let-7i, this was indicated at 103 copies/µL with a sensitivity of 43% and a specificity of 14% (AUC = 0.64, *p* < 0.05, 95% confidence interval = 0.513–0.774). Combinations of all tested miRNAs have been made, however the results have not been improved. For details, see Appendix A.

## 3. Discussion

The let-7 (lethal-7) miRNA was one of the first miRNAs described in the nematode *Caenorhabditis elegans*, and its biological functions show a high level of evolutionary conservation from the nematode to the human. Higher animals have multiple isoforms of let-7 miRNAs, unlike in *C. elegans*. The human let-7 miRNA family contains nine individual mature let-7 miRNAs (let-7a, let-7b, let-7c, let-7d, let-7e, let-7f, let-7g, let-7i, and miR-98) with identical seed sequences. All let-7 miRNAs are produced from 12 genes (including let-7-a1, -a2, -a3, -b, -c, -d, -e, -f1, -f2, -g, -i, and miR-98) located on eight different chromosomes (3, 9, 11, 12, 19, 21, 22, and chromosome X) [12].

In the present study, we have analyzed five members of the let-7 family. We have noticed significantly increased levels of let-7a, let-7c, let-7d, and let-7f in plasma of PTC patients compared to healthy controls. Noteworthily, to our knowledge, there are currently no studies that have examined expression of these miRNAs in the plasma of PTC patients. Additionally, let-7i showed an increase in expression of PTC plasma samples of borderline significance.

The expression of let-7 is different in different cancer types. In breast, colon, prostate and renal cancer, Heneghan et al. [13] observed a significant increase of let-7a in the blood of cancer patient samples compared to healthy controls. Similarly, Ogata-Kawata et al. [14] detected significantly higher level of let-7a in serum exosomes from colon cancer patients than in those from healthy controls. In contrast to this, Kelly et al. [15] detected decreased expression of let-7a in the whole blood of prostate cancer samples. Moreover, Tsujiura et al. [16] also identified significantly reduced level of let-7a in plasma and tissue of gastric cancer patients compared to healthy controls. In colorectal cancer, Ghanbari et al. [17] observed a decrease in let-7a expression in plasma samples. Additionally, in plasma and tissue samples of breast cancer there was observed a decreased level of let-7a [18]. In the serum of the breast cancer patients, Li et al. [19] observed significantly decreased level of let-7c compared with the healthy controls. On the other hand, elevated plasma levels and decreased tissue levels of let-7c were observed by Qattan et al. [20] in breast cancer patients. Thus, the differences in expression of let-7c in breast cancer may result from the use of different types of material (serum vs. plasma). In cervical cancer, Zhang et al. [21] showed the significantly decreased level of let-7d between tumors and adjacent normal tissues. Moreover, they presented a consistent trend in plasma samples. The level of let-7f was significantly elevated in the serum of gastric cancer patients in the study by Liu et al. [22]. On the other hand, Ghanbari et al. [17] observed a decrease in let-7f expression in plasma samples of colorectal cancer. Elevated level of let-7i was observed in plasma of breast cancer patients and in serum of gastric cancer patients [20,22].

The discrepancy in results may be due to the different nature of various types of cancer. Differences also occur within the same type of cancer and analyzed miRNA, which may result from the use of different methodology and the material used (whole blood, plasma, serum). An example of these are results of Li et al. and Qattan et al. Both authors analyzed let-7c in breast cancer, but Li used serum and Qattan used plasma. Additionally, these highly variable results may be due to a variety of patient variables such as age, cancer stage, and controls used [19,20].

Regarding the analyzed let-7 family members in PTC, let-7a was increased in the tissue compared to control [23]. Contradictory results were obtained for let-7b. An increase was observed both in plasma [24] and PTC tumors sample [25], whereas Li et al. [26] detected decreased expression of let-7b in PTC tumor samples. Decreased expression in PTC tumors was found for let-7d [27,28]. Upregulated expression of let-7e in plasma of PTC patients was observed. Furthermore, there was no significant difference in the levels of let-7e expression between the PTC and benign nodule tissues [29]. Contradictory results were obtained for the expression of let-7f in tissue. Pallante et al. [30] observed decreased expression of let-7f in PTC tumors, while Damanakis et al. [25] also identified overexpression in PTC tumor samples. For let-7g, downregulated expression was found in tissue samples [27]. Similar to let-7e, let-7i showed an increase in plasma of PTC patient samples, and no significant difference in the levels of this miRNA expression was found in tissue compared to control [31]. In this work, upregulated expression of those miRNA in plasma of PTC patient samples was observed with borderline significance (*p* = 0.042). Similarly, in our work, for let-7i, the difference between the groups reached borderline significance (*p* = 0.059).

Despite all let-7 family members having similar structure, let-7i showed the greatest discrepancy in the base sequence against let-7a [32]. This may cause different expression levels of the individual let-7 members.

On the other hand, a similar structure may indicate a similar function of all let-7 miRNAs, which resulted in the elevated level of all examined let-7 members but at different levels. It has been established on hepatoma cells that when let-7i and let-7g were overexpressed together, it had a greater impact on division and apoptosis than overexpression of separate miRNAs, suggesting that members of this family may act in cooperation [33]. However, there is some evidence that, probably due to unique target preferences, different members of the let-7 family do have different roles, and thus cannot be considered as one. It has been verified that overexpression of different let-7 family members affects cell viability on different levels in hepatocellular carcinoma [34]. Our ROC analysis has shown that structurally related let-7c and let-7f have similar sensitivity and specificity, as do let-7a and let-7d. Moreover, sensitivity for let-7a and let-7d above 70% indicate these miRNAs to be a useful tool in diagnosis of PTC. High level of those miRNAs in patient’s plasma may confirm the presence of PTC in case when FNB results are inconclusive.

Abnormal expression of let-7 has been associated with cancer initiation and progression. Substantial evidence suggests that let-7 could function as a tumor suppressor or an oncogene through various stimuli.

One of the explanations can be that cancer cells release the intercellular tumor-suppressive miRNAs to the extracellular environment, change tumor microenvironment and support cancer progression. Circulating miRNAs can directly target tissue mRNAs of genes involved in the cell immunity, differentiation, or angiogenesis and consequently may lead to carcinogenesis. Tumor-associated macrophages (TAMs) are a class of immune cells present in high numbers in the microenvironment of solid tumors. Baer et al. demonstrated that increased let-7 expression in TAMs results in the change of M1 into the M2 phenotype. The M2 phenotype is associated with increased angiogenesis and tumor aggressiveness, while TAMs with M1 phenotype show proinflammatory activity and better prognosis [35]. Moreover, Qattan et al. also suggest that breast cancer tumor cells selectively secrete tumor suppressor miRNAs to maintain oncogenesis [20].

On the other hand, the phenomenon that tumor-suppressing miRNAs were increased reflected the fact that circulating miRNAs were not the main products of cancerous cells but the results of overall immune response and play a significant role in cancer defense and cancer therapy. We have still only begun to understand the complexity of let-7 in immune system regulation. Several reports suggest that let-7 is essential for CD8+ cells maturation, and its expression significantly decreases in late stages. However, the exact mechanism of let-7-mediated regulation is unclear. Pobezinky et al. have shown that the naive CD 8 cells with lin28-mediated knockdown of let-7 expression revealed comprised hemostasis of this cell in vivo, which finally leads to increased proliferation of these cells [36]. This might increase cytotoxic T cells and natural killer cytotoxicity, which plays a crucial role in killing the cancer cells and eradicating the tumor. This is not surprising in the context that PTC presents very limited metastatic abilities and is a relatively nonaggressive tumor.

Next to the diagnostic opportunity, circulating let-7 can be used for therapy monitoring purposes. This requires an individual approach in each type of cancer. Let-7 levels are directly affected by the therapy. For instance, in acute promyelocytic leukemia, let-7c increases rapidly after chemotherapy, then decreases again upon relapse [37] Moreover, monitoring of let-7 levels can allow treatment with a lower dose of chemotherapy to obtain the same therapeutic benefit. This represents an opportunity to avoid severe side-effects of cancer treatments by using lower chemotherapy dosages. Acute myeloid leukemia, lung cancers, and chemoresistant epithelial ovarian cancers are characterized by a reduced let-7 resulting in nonresponse to chemotherapy [38,39,40,41,42,43]. Interestingly, decreased expression of let-7 was associated with significantly shorter survival after resection and may have a prognostic impact on the survival of surgically-treated lung cancer patients [44]. These are examples of methods in which monitoring let-7 levels could be used to predict drug response or recurrence.

All of these show the complexity of the connection between let-7 and cancer development. Thus, it is important that let-7 overexpression treatment strategies be personalized concerning individualized clinical approaches based on specific miRNA expression profiles, as opposed to overarching treatment schemes spanning across multiple malignancy types. However, this will require careful consideration and further retrospective trials followed by robust clinical trials [45].

## 4. Materials and Methods

### 4.1. Patients and Blood Samples

The level of miRNAs in plasma was evaluated in patients admitted to the Department of General and Oncological Surgery of the Medical University of Lodz. The patients were suspected of, or diagnosed with, papillary thyroid cancer based on fine needle aspiration biopsy (FNAB) before the surgery. The surgical procedure was performed on all patients in accordance with the guidelines of the Polish Society of Surgeons and the Polish Society of Oncological Surgery [46]. A total of 49 patients (44 females and 5 males) with histologically confirmed PTC was enrolled. The inclusion and exclusion criteria were previously described [4]. Staging of the tumors was carried out according to the system approved by the International Union Against Cancer (UICC, 2010) and/or WHO classification. The 21 control subjects (12 females and 9 males) were healthy volunteers. For details, see Appendix A. All patients and volunteers provided written informed consent. The protocol was reviewed and approved by The Medical University of Lodz Ethics Committee (RNN/146/18/KE dated 15 May, 2018) and in compliance with the Declaration of Helsinki.

### 4.2. Blood Sample Processing and miRNA Extraction

For total RNA extraction and analysis for miRNAs using ddPCR analysis, 9 mL venous blood was collected from each patient in vacutainer tubes (Becton Dickinson, Franklin Lakes, NJ) with EDTA. The samples were collected before surgery and processed within 2 h after venipuncture. Plasma from EDTA blood samples was obtained by centrifugation (1600× *g*, 4 °C) for 10 min. Then, plasma samples were brought to the second centrifugation (16,000× *g*, 4 °C, 5 min) to remove the cell debris and were stored at −80 °C for no longer than 4 weeks until miRNAs level measurements were obtained. Total RNA including miRNAs were extracted from 1 mL of plasma using QIAamp ccfDNA/RNA Kit (Cat No./ID: 55184) according to the manufacturer’s instruction. RNA was eluted in 20 μL of nuclease-free water. An equal volume of starting material was used for all the plasma samples to adjust for differences in recovery efficiency between different plasma samples.

Hemolysis can potentially affect the accuracy of miRNA quantification in a blood sample. Therefore, plasma samples collection and preparation for miRNA evaluation were followed by the SOPs (Standard Operating Procedures) [47]. Visibly hemolyzed samples (pink to red discoloration of the plasma) were excluded from analysis. In addition, the low-grade hemolysis was assessed by Picodrop Spectrophotometer. Samples were classified as hemolyzed or not hemolyzed based on the optical density of free hemoglobin at 414 nm, as previously reported [48]. Absorbance at 414 nm of higher than 0.2 was used as a cutoff to distinguish hemolyzed and nonhemolyzed plasma. The absorbance values of all samples were below 0.2.

### 4.3. Reverse Transcription

The miRNAs were reverse-transcribed using the TaqMan miRNA Reverse Transcription Kit (#4366596) and the associated miRNA-specific stem-loop primers for five miRNAs: let-7a, let-7c, let-7d, let-7f, and let-7i (Applied BioSystems, Foster City, CA; Cat. No. 4440887, assay ID 000377, assay ID 002283, assay ID 000382, assay ID 000379, assay ID 000384) with some modifications. A customized RT primer pool was prepared by pooling all miRNA-specific stem-loop primers of interest. Six microliters of this mixture were added to the reaction mix containing 5 μL of extracted total RNA, 0.3 μL 100 mM dNTP, 3 μL enzyme (50 U/μL), 1.5 μL 10× RT buffer, and 0.19 μL RNase inhibitor (20 U/μL). A final volume of 15 μL was reverse-transcribed according to the manufacturer’s instructions. cDNA was then stored at −20 °C. In order to normalized the sample-to-sample variability in miRNA content, a fixed volume of eluted RNA sample was used as input, rather than the same amount of RNA for each sample. Indeed, the yield of RNA from a small volume of plasma/serum has been reported to be below the limit of accurate quantitation by spectrophotometry, thus miRNA contents extracted from plasma/serum has been reported to be undetectable by using NanoDrop spectrophotometer.

### 4.4. Digital Droplet PCR

Total plasma levels of let-7a, let-7c, let-7d, let-7f, and let-7i were quantified using the ddPCR system (Bio-Rad Laboratories) in Central Scientific Laboratory “CoreLab” of the Medical University of Lodz. Briefly, 1 μL of the synthesized cDNA was added to a 19 μL PCR reaction mixture containing 10 μL of digital PCR ™ supermix (Bio-Rad Laboratories), 1 μL of TaqMan primer/probe mix (Applied BioSystems) and RNase-free H_2_O. Droplets were generated by loading the mixture into a single-use cartridge with 70 μL of QX100 Droplet Generation oil (Bio-Rad Laboratories). Cartridges were then placed into the QX200 Droplet Generator (Bio-Rad Laboratories). The droplets generated from each sample were transferred to a 96-well PCR plate (Eppendorf, Hamburg, Germany), and PCR amplification was carried out on the C1000 Touch Thermal Cycler (Bio-Rad Laboratories), according to the manufacturer’s protocol. The thermal cycling conditions were as follows: 95 °C for 10 min, 40 cycles of 95 °C for 15 s and 60 °C for 1 min, and a final inactivation step at 98 °C for 10 min. NTCs were included in every PCR plate. The plate was then loaded on the QX200 Droplet Reader (Bio-Rad Laboratories) and read automatically. The fraction of PCR-positive droplets was quantified assuming a Poisson distribution [49]. In detail, the QuantaSoft software was used to obtain the concentration results in number of copies per microliter (copies/μL) for each sample.

### 4.5. Statistical Analysis

All statistical analyses were performed using Dell Statistica (data analysis software system), version 13 (software.dell.com). The Shapiro–Wilk test showed that our data were not normally distributed. The nonparametric Mann–Whitney test was used for comparison between two groups. The Kruskal–Wallis test (nonparametric ANOVA) was used for comparison between the three groups. For the prediction of cutoff values of the markers studied, the receiver operating characteristic (ROC) curve have been used. All *p*-values ≤ 0.05 were considered statistically significant. Numerical data were expressed as mean ± SD (median).

## Figures and Tables

**Figure 1 diagnostics-10-00130-f001:**
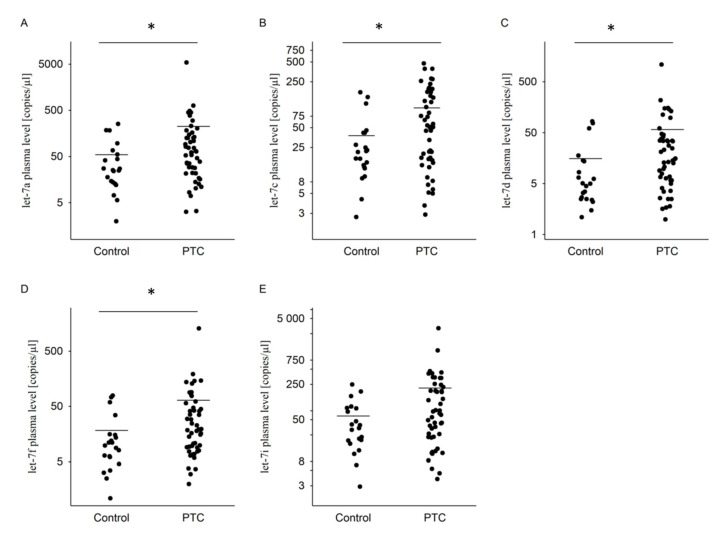
The dot plots of expression of the selected miRNAs in plasma of the PTC patients compared to healthy control, showing copies/µL of (**A**) let-7a, (**B**) let-7c, (**C**) let-7d, (**D**) let-7f, and (**E**) let-7i. The line represents the mean value. * *p* ≤ 0.05. Statistically significant differences were determined by Mann–Whitney test.

**Figure 2 diagnostics-10-00130-f002:**
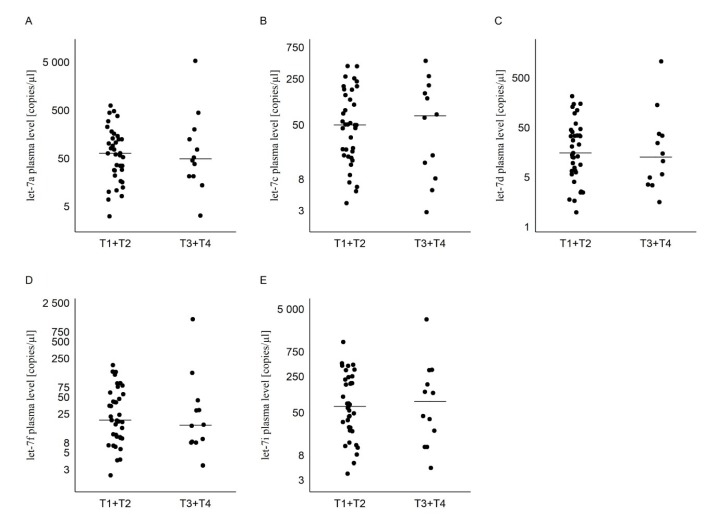
The dot plots of expression of the selected miRNAs in plasma of the PTC patients, comparing T_1_ + T_2_ and T_3_ + T_4_ and showing copies/µL of (**A**) let-7a, (**B**) let-7c, (**C**) let-7d, (**D**) let-7f, and (**E**) let-7i. The line represents the mean value.

**Figure 3 diagnostics-10-00130-f003:**
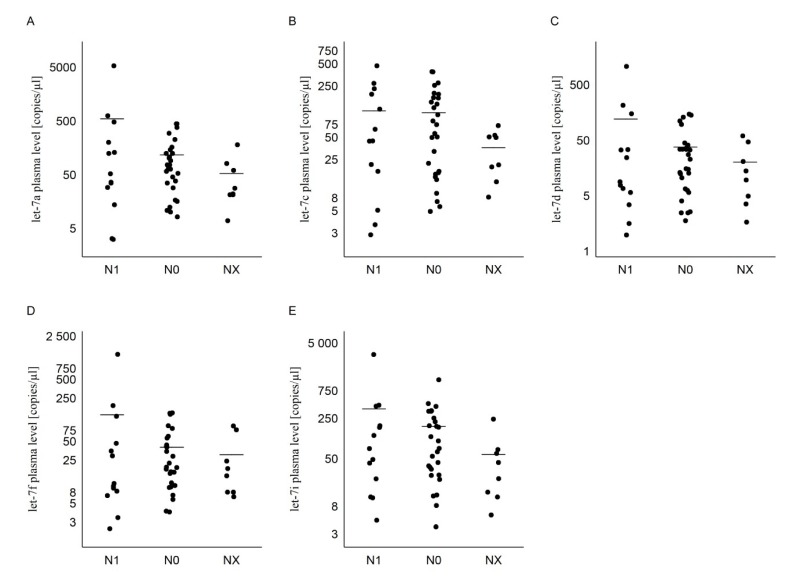
The dot plots of expression of the selected miRNAs in plasma of the PTC patients, comparing N_1_, N_0_, and N_X_ and showing copies/µL of (**A**) let-7a, (**B**) let-7c, (**C**) let-7d, (**D**) let-7f, and (**E**) let-7i. The line represents the mean value.

**Figure 4 diagnostics-10-00130-f004:**
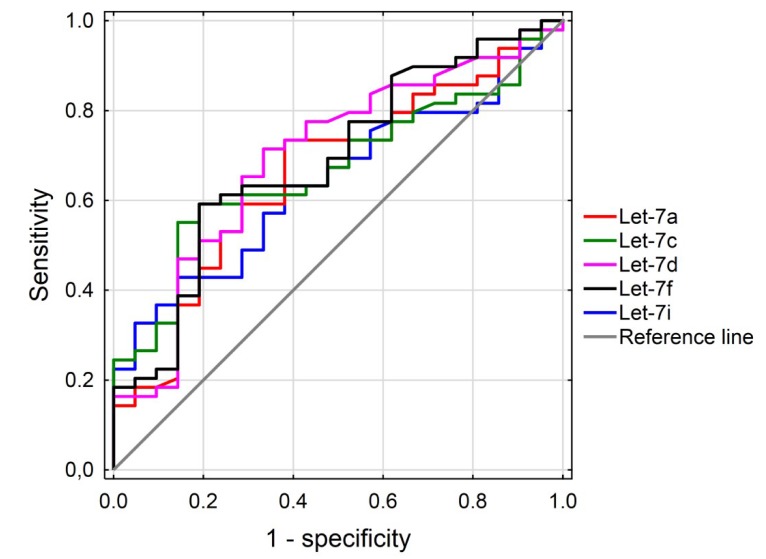
Receiver-operator characteristic (ROC) curves considering all the samples from the training data set (49 cancer cases and 21 controls) for the let-7a, let-7c, let-7d, let-7f, and let-7i miRNAs. The ROC curves plot sensitivity versus 1−specificity. The determined cutoff values for miRNAs were 28.5 copies/µL, 50.5 copies/µL, 6.8 copies/µL, 16.2 copies/µL, and 103 copies/µL, respectively.

**Table 1 diagnostics-10-00130-t001:** Demographics of the study participants (clinical characteristics).

Variability	Control	PTC
**Gender**	21	49
Male	9 (43%)	5 (10%)
Female	12 (57%)	44 (90%)
**Age (years)**		
Median	47	48
Range	29–67	20–76
**T&N staging**		
T_1_ + T_2_	—	37 (76%)
T_3_ + T_4_	—	12 (24%)
N_0_	—	28 (57%)
N_1_	—	13 (27%)
N_X_	—	8 (16%)

**Table 2 diagnostics-10-00130-t002:** Level of miRNAs in study groups.

miRNAs	Control (Copies/µL) *n* = 21	PTC (Copies/µL) *n* = 49
**Let-7a**	55 ± 69 (25)	224 ± 776 (62) *
**Let-7c**	38 ± 47 (21)	100 ± 115 (51) *
**Let-7d**	16 ± 25 (5)	58 ± 159 (15) *
**Let-7f**	18 ± 23 (10)	65 ± 185 (19) *
**Let-7i**	59 ± 64 (34)	212 ± 483 (67)

The data are expressed as mean ± standard deviation (median). * Statistically significant data corresponding control, *p* ≤ 0.05. Statistically significant differences were determined by Mann–Whitney test. PTC: papillary thyroid carcinoma.

**Table 3 diagnostics-10-00130-t003:** Comparison of miRNAs levels and gender, the primary tumor (T) and regional lymph nodes (N).

**PTC (Copies/µL)**					
**Variability**	**Let-7a**	**Let-7c**	**Let-7d**	**Let-7f**	**Let-7i**
**Gender**					
Male	38 ± 17 (37)	32 ± 27(20)	10 ± 7 (9)	13 ± 7 (10)	48 ± 27 (44)
Female	245 ± 817 (72)	107 ± 119 (53)	63 ± 167 (19)	71 ± 194 (21)	230 ± 507 (72)
**T&N staging**					
T_1_ + T_2_	121 ± 150 (65)	93 ± 106 (51)	39 ± 52 (16)	42 ± 47 (20)	160 ± 222 (67)
T_3_ + T_4_	543 ± 1 551 (50)	119 ± 143 (69)	115 ± 311 (13)	134 ± 367 (16)	372 ± 905 (83)
N_0_	119 ± 130 (72)	110 ± 114 (67)	38 ± 45 (19)	41 ± 43 (21)	183 ± 240 (90)
N_1_	555 ± 1 484 (53)	116 ± 142 (45)	123 ± 300 (9)	137 ± 352 (11)	367 ± 869 (76)
N_X_	54 ± 59 (25)	36 ± 24 (36)	20 ± 22 (12)	31 ± 33 (16)	60 ± 79 (33)
**Control(Copies/µL)**					
	**Let-7a**	**Let-7c**	**Let-7d**	**Let-7f**	**Let-7i**
**Gender**					
Male	30 ± 28 (25)	18 ± 13 (21)	5 ± 4 (5)	11 ± 10 (10)	30 ± 22 (32)
Female	73 ± 86 (33)	53 ± 58 (20)	24 ± 31 (7)	24 ± 29(10)	81 ± 77 (61)

Data are expressed as mean ± standard deviation (median).

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
