# Peer review of "Analysis of Let-7 Family miRNA in Plasma as Potential Predictive Biomarkers of Diagnosis for Papillary Thyroid Cancer"

_diagnostics, 2020, doi:10.3390/diagnostics10030130_

Round 1
Reviewer 1 Report
The manuscript "Analysis of let-7 Family miRNAs in Plasma as Potential Predictive Biomarkers of Diagnosis for Papillary Thyroid Cancer" presented by E. Perdas et al., was aimed to evaluate the diagnostic and prognostic significance of circulating let7 Family miRNA levels for the management of papillary thyroid cancer (PTC). In this study, relative levels of five miRNAs (let-7a, let-7c, let-7d, let-7f, and let-7i) have been examined in the plasma of patients with PTC, and compared them with those of a healthy control group. Moreover, correlations of the expression levels of let-7 Family miRNAs with gender and age characteristics, as well as clinico-pathological (PTC stages) characteristics of patient cohorts were investigated. Importantly, in this study droplet digital PCR allowing absolute quantification without the need for internal normalization of circulating miR expression was used to improve accuracy and reproducibility of results. To calculate the diagnostic value of let-7a, let-7c, let-7d, let-7f miRNAs in the PTC, the Receiver operating characteristic (ROC) curve analyses were performed for estimating the specificity and sensitivity of miRNA to diagnose PTC patients.
In this study, no statistical difference in mRNAs levels were found among men and women in the PTC group compared to controls. Similarly, there was no statistical difference in miRNA levels compared to T1+T2 and T3+T4 tumor stages, no statistical difference was found between the group of patients with regional lymph node metastases (N1) compared to no regional lymph nodes metastases (N0) and group where lymph nodes could not be assessed (Nx) for all miRNAs. However, the most significant outcome of the study is finding of up-regulated expression of let-7a, let-7c, let-7d and let-7f in PTC patients compared to healthy control groups.
This work provides an advance towards the current knowledge on let-7 Family miRNA expression in PTC and their possible diagnostic significance.These findings may also have value for treatment prognosis for the patients with PTC. In my opinion, this clinical study has adequate design and high quality of presentation. The novelty and significance are sufficiently substantiated in the Introduction and Discussion. Methods are fully described. Description of results is detailed and complete. The analysis of miRNA expression and statistical data processing were carried out accurately, with all the required steps. List of references is reasonably sufficient and includes all relevant publications. The conclusions are supported by the results. In the Discussion section, the authors carefully and comprehensively discuss the results of previous studies and compare with their own data. They take into account multiple contradictory and consistent data on let-7 Family miRNA expression in different types of cancer and when using different methods of analysis of different samples (tumor, blood and plasma samples). They also discuss the possible mechanisms of let-7-mediated regulation of cancer progression and involvement of immune cells. In my opinion, the present work is of interest to a broad audience of clinicians and basic researchers.
I have the following minor comments:
- In the Discussion section, the authors use the terms "PTC tissues" and "PTC plasma" (P5, lines: 165-177). I think that the use of such terms is incorrect. Undoubtedly, they need to correct "PTC plasma" to "plasma of PTC patient". And "PTC tissues" can be replaced with "PTC cells" or "PTC tumors" depending on the context.
- Sentence "Upregulated expression of PTC plasma was observed of let-7e" needs to be corrected (P5, Line 169).
- The last sentence of abstract seems inefficient ("However, this will require careful investigation and further retrospective trials followed by robust clinical trials") and also needs to be edited.
Author Response
Response to Reviewer #1 Comments
To reviewer: “First of all, we would like to express our appreciation for your in-depth comments, suggestions, and corrections, which have greatly improved the manuscript”
The manuscript "Analysis of let-7 Family miRNAs in Plasma as Potential Predictive Biomarkers of Diagnosis for Papillary Thyroid Cancer" presented by E. Perdas et al., was aimed to evaluate the diagnostic and prognostic significance of circulating let7 Family miRNA levels for the management of papillary thyroid cancer (PTC). In this study, relative levels of five miRNAs (let-7a, let-7c, let-7d, let-7f, and let-7i) have been examined in the plasma of patients with PTC, and compared them with those of a healthy control group. Moreover, correlations of the expression levels of let-7 Family miRNAs with gender and age characteristics, as well as clinico-pathological (PTC stages) characteristics of patient cohorts were investigated. Importantly, in this study droplet digital PCR allowing absolute quantification without the need for internal normalization of circulating miR expression was used to improve accuracy and reproducibility of results. To calculate the diagnostic value of let-7a, let-7c, let-7d, let-7f miRNAs in the PTC, the Receiver operating characteristic (ROC) curve analyses were performed for estimating the specificity and sensitivity of miRNA to diagnose PTC patients.
In this study, no statistical difference in mRNAs levels were found among men and women in the PTC group compared to controls. Similarly, there was no statistical difference in miRNA levels compared to T1+T2 and T3+T4 tumor stages, no statistical difference was found between the group of patients with regional lymph node metastases (N1) compared to no regional lymph nodes metastases (N0) and group where lymph nodes could not be assessed (Nx) for all miRNAs. However, the most significant outcome of the study is finding of up-regulated expression of let-7a, let-7c, let-7d and let-7f in PTC patients compared to healthy control groups.
This work provides an advance towards the current knowledge on let-7 Family miRNA expression in PTC and their possible diagnostic significance. These findings may also have value for treatment prognosis for the patients with PTC. In my opinion, this clinical study has adequate design and high quality of presentation. The novelty and significance are sufficiently substantiated in the Introduction and Discussion. Methods are fully described. Description of results is detailed and complete. The analysis of miRNA expression and statistical data processing were carried out accurately, with all the required steps. List of references is reasonably sufficient and includes all relevant publications. The conclusions are supported by the results. In the Discussion section, the authors carefully and comprehensively discuss the results of previous studies and compare with their own data. They take into account multiple contradictory and consistent data on let-7 Family miRNA expression in different types of cancer and when using different methods of analysis of different samples (tumor, blood and plasma samples). They also discuss the possible mechanisms of let-7-mediated regulation of cancer progression and involvement of immune cells. In my opinion, the present work is of interest to a broad audience of clinicians and basic researchers.
I have the following minor comments:
Point 1: In the Discussion section, the authors use the terms "PTC tissues" and "PTC plasma" (P5, lines: 165-177). I think that the use of such terms is incorrect. Undoubtedly, they need to correct "PTC plasma" to "plasma of PTC patient". And "PTC tissues" can be replaced with "PTC cells" or "PTC tumors" depending on the context.
Response 1: We thank the Reviewer for providing such a thorough review of our paper. All terms "PTC tissues" and "PTC plasma" were checked and corrected according to reviewer suggestion (line: 188-197).
Point 2: Sentence "Upregulated expression of PTC plasma was observed of let-7e" needs to be corrected (P5, Line 169).
Response 2: Sentence has been corrected (line: 189). ”Upregulated expression of let-7e in plasma of PTC patients was observed.”
Point 3: The last sentence of abstract seems inefficient ("However, this will require careful investigation and further retrospective trials followed by robust clinical trials") and also needs to be edited.
Response 3: Sentence has been corrected (line:24-25).” However, our observation requires further research on a larger patient group.”
To Editor and Reviewer:
“Again, we would like to thank you for all the consideration that you gave us for this manuscript.”
Reviewer 2 Report
The authors have attempted to characterise let-7 family miRNA in plasma of PTC patients as potential biomarkers of diagnosis.
Introduction:
Line 63: “overwhelmed”. The authors mean overcome?
Results:
Table 2: It would be good to represent the data shown in Table 2 as a dot plot to show the distribution of the expression of Let7 miRNA family. This would help highlight lack of significance for Let7i.
Line 85: “… level of let-7i was insignificance higher…” Awkward sentence.
The authors have attempted to show that there was no statistical difference between the gender and expression of let7 miRNA family, however, lack of significance can be mainly due to small numbers in the male cohort especially in the PTC group (n=5). Once again showing Table 3 as dot plots will highlight the reason for insignificance and the comparisons being made.
Line 89: what statistical analysis was carried out? Also was it one-tailed or two-tailed.
Line 100: no statistical difference between T1+T2 vs T3+T4? Once again showing this as dot plots highlighting the spread of the data will help understand the lack of significance. Also was a one-tailed or two-tailed test carried out?
Table 3: Can the authors check the values for mean +/- Sd(median) for all the Let7 family especially for Let-7a and 7i for T1+T2 and N1? It does not seem correct?
Why have the authors used Kruskal-Wallis test over Mann-whitney test for the lymph node status in Table 3?
Figure 1: Why have the authors not tested Let-7i?
Also the authors should have tried to see if combinations of the different Let miRNA improved sensitivity and specificity of the test. What is used in the clinic to assess prognosis for PTC? If staging is used or lymph node involvement, how does one or more of the let-7 miRNA family in the plasma improve prediction of outcome for PTC patients. These analysis need to be added, else it undermines the importance of this work.
Discussion:
Line 138-139: “Additionally, Let-7i…” – sentence is not correct?
Line 141: blood of cancer samples: I presume the authors means blood of cancer patients compared to…
There are so many documents of Let7 miRNAs being elevated in various cancer types, how can these be used solely as diagnostic biomarkers for PTC? This needs to be added to the discussion. The authors have provided a list of cancers where this family of miRNAs has been characterised but failed to highlight how their findings are specific for PTC. This needs to be addressed in the discussion section.
Method:
The authors have classified the samples into hemolyzed and not, however, they have not presented a comparison of the Let7 miRNA RNAs due to the presence of hemolysis. This would add lots of value to their study.
Author Response
Response to Reviewer #2 Comments
To reviewer: “First of all, we would like to express our appreciation for your in-depth comments, suggestions, and corrections, which have greatly improved the manuscript”
The authors have attempted to characterise let-7 family miRNA in plasma of PTC patients as potential biomarkers of diagnosis.
Introduction:
Point 1: Line 63: “overwhelmed”. The authors mean overcome?
We have mean “overcome”. This have been corrected (line: 64).
Results:
Point 2: Table 2: It would be good to represent the data shown in Table 2 as a dot plot to show the distribution of the expression of Let7 miRNA family. This would help highlight lack of significance for Let7i.
Dot plots have been added to the manuscript (Figure. 1).
Point 3: Line 85: “… level of let-7i was insignificance higher…” Awkward sentence.
Sentence has been corrected (line:85-87). “The level of let-7i was higher in PTC group compare to control but the difference is statistically insignificant (Mann-Whitney: p=0.059).”
Point 4: The authors have attempted to show that there was no statistical difference between the gender and expression of let7 miRNA family, however, lack of significance can be mainly due to small numbers in the male cohort especially in the PTC group (n=5). Once again showing Table 3 as dot plots will highlight the reason for insignificance and the comparisons being made.
We agree that lack of significance between gender may be due to small numbers in the male cohort in the PTC group but PTC affected much more often women than men [1]. For this reason, adding charts seems less clear and the table seems sufficient.
Point 5: Line 89: what statistical analysis was carried out? Also was it one-tailed or two-tailed.
Two-tailed Mann-Whitney test was carried out. The type of used test has been added (line 90-91).
Point 6: Line 100: no statistical difference between T1+T2 vs T3+T4? Once again showing this as dot plots highlighting the spread of the data will help understand the lack of significance. Also was a one-tailed or two-tailed test carried out?
Dot plots have been added to the manuscript (Figure. 2). Two-tailed test was carried out. Moreover, N1 vs N0 vs NX dot plots have been added too (Figure. 3).
Point 7: Table 3: Can the authors check the values for mean +/- Sd(median) for all the Let7 family especially for Let-7a and 7i for T1+T2 and N1? It does not seem correct?
We have verify all the values, however no mistakes have had been found.
Point 8: Why have the authors used Kruskal-Wallis test over Mann-whitney test for the lymph node status in Table 3?
The Kruskal-Wallis test is used to compare more than two variables. In our case to compare N1, N2, and NX.
Point 9: Figure 1: Why have the authors not tested Let-7i?
Let-7i have been added to Figure 4 and all ROC data for this miRNA have been added in text (line: 133;135; 143-144).
Point 10: Also the authors should have tried to see if combinations of the different Let miRNA improved sensitivity and specificity of the test. What is used in the clinic to assess prognosis for PTC? If staging is used or lymph node involvement, how does one or more of the let-7 miRNA family in the plasma improve prediction of outcome for PTC patients. These analysis need to be added, else it undermines the importance of this work.
and we decided not add it to manuscript. Below combined results:
|
Let-7 |
Sensitivity |
Specificity |
|
Let-7a+ Let-7c |
53 |
14 |
|
Let-7a+ Let-7f |
59 |
19 |
|
Let-7a+ Let-7d |
71 |
33 |
|
Let-7c+ Let-7f |
65 |
28 |
|
Let-7c+ Let-7d |
55 |
14 |
|
Let-7f+ Let-7d |
59 |
23 |
We add this table and info about combined results to the supplementary file and manuscript (line: 144-146)
Discussion:
Point 11: Line 138-139: “Additionally, Let-7i…” – sentence is not correct?
Sentence have been corrected (line:158). “Additionally, let-7i showed increase in expression of PTC plasma samples of borderline significance.”
Point 12: Line 141: blood of cancer samples: I presume the authors means blood of cancer patients compared to…
Sentence have been corrected (line: 160-162). “ In breast, colon, prostate and renal cancer, Heneghan et al. [13] observed a significant increase of let-7a in the blood of cancer patient samples compared to healthy controls.”
Point 13: There are so many documents of Let7 miRNAs being elevated in various cancer types, how can these be used solely as diagnostic biomarkers for PTC? This needs to be added to the discussion. The authors have provided a list of cancers where this family of miRNAs has been characterised but failed to highlight how their findings are specific for PTC. This needs to be addressed in the discussion section.
Potential usage of our findings to diagnosis of PTC has been added in discussion section (line: 211-213). “Moreover, sensitivity for let-7a and let-7d above 70% indicate these miRNA to be a useful tool in diagnosis of PTC. High level of those miRNA in patient’s plasma may confirm the presence of PTC in case when FNB results are inconclusive.”
Method:
Point 14: The authors have classified the samples into hemolyzed and not, however, they have not presented a comparison of the Let7 miRNA RNAs due to the presence of hemolysis. This would add lots of value to their study.
Hemolyzed samples were not used for analysis. Hemolysis can potentially affect the accuracy of miRNA quantification in a blood sample. Visibly hemolyzed samples (pink to red discoloration of the plasma) were excluded from analysis. In addition, the low-grade hemolysis was assessed by Picodrop Spectrophotometer according to Kirschner et. al.[2].
Bibliography
- Alok Pathak, K.; Leslie, W.D.; Klonisch, T.C.; Nason, R.W. The changing face of thyroid cancer in a population-based cohort. Cancer Med. 2013, 2, 537–544.
- Kirschner, M.B.; Kao, S.C.; Edelman, J.J.; Armstrong, N.J.; Vallely, M.P.; van Zandwijk, N.; Reid, G. Haemolysis during Sample Preparation Alters microRNA Content of Plasma. PLoS One 2011, 6, e24145.
To Editor and Reviewer:
“Thank you very much for these constructive comments and suggestions. They significantly improved quality of our manuscript.”
Round 2
Reviewer 2 Report
The authors have addressed all concerns raised and the article is good to be published.